
# A mathematical formulation for estimating maximum run-up height of 2018 Palu tsunami

Ikha Magdalena[1], Antonio Hugo Respati Dewabrata[1], Alvedian Mauditra Aulia Matin[1], Adeline Clarissa[1], and Muhammad Alif Aqsha[1]

[1]Industrial and Financial Mathematics Group, Institut Teknologi Bandung, West Java, Indonesia

**Correspondence:** Ikha Magdalena (ikha.magdalena@math.itb.ac.id)

**Abstract.** Run-up is defined as sea wave up-rush on a beach. Run-up height is affected by many factors, including the shape of the bay. As an archipelagic country, Indonesia consists of thousands of islands with bays of diverse profiles, including Palu Bay, which is a well-known example of a bay with a drastically-increasing wave run-up height. In the case of the 2018 Palu tsunami, scientists found that the incident wave was amplified by the shape of the bay. The amplifying wave played a large
role in the significant increase of run-up height. The run-up in question caused severe inundation, which led to a high number of casualties and damages. Therefore a mathematical model will be constructed to investigate the wave run-up. The bay's geometry will be approximated using three linearly-inclined channel types: one of parabolic cross-section, one of triangular cross-section, and a plane beach. We use the generalized nonlinear shallow water equations, which is then solved analytically using a hodograph-type transformation. As a result, the nonlinear shallow water equation system can be reduced to a one-
dimensional linear equation system. Assuming the incident wave is sinusoidal, we can obtain a simple formula for calculating maximum run-up height on the shoreline.

## 1 Introduction

Indonesia is an archipelagic country consisting of at least 17,508 islands with over 80,000 kilometers of convoluted coastline. This enormous number makes Indonesia the country with the fourth longest coastline in the world, according to the World
Resource Institute (Wood et al., 2000). As such, many bays of diverse shapes can be found along the coastline. Analysis of the characteristics of waves in diversely shaped bays is a major area of interest for researchers (Didenkulova and Pelinovsky, 2011a; Didenkulova, 2013; Rybkin et al., 2014). Previous research has proven that the shape of a bay has a significant impact on run-up heights of waves that propagate within it. Longer, narrower bays will generate run-ups of greater height compared to inclined plane beaches.

This natural phenomenon is relevant to the circumstances of most bays in Indonesia, such as Balikpapan Bay, Semangka Bay, and Palu Bay. Indonesia is located between two continental plates, the Australian and Sunda plates, as well as two oceanic plates, the Philippine and Pacific Sea plates, which increases the risk of devastating tsunamis occurring near these bays. The 2018 Palu tsunami is evidence of how severe the effect of the shape of the bay is. Researchers have considered that Palu Bay's long and narrow physical shape plays a large role in amplifying the incident wave by squeezing the wave in question from its





sides (Socquet et al., 2019). On the 28th of September, 2018, a 7.5 $M_w$ magnitude earthquake struck the Donggala regency. Accordingly, a tsunami warning was issued in Palu. It was predicted that tsunami waves less than 3 meters tall would hit the city. However, the actual recorded tsunami wave heights were about 4-5 meters. This unanticipated calamity attracted the attention of some scientists and researchers who wished to investigate further. One such group theorized that the increase in tsunami height was mostly caused by coastal and submarine landslides, as characterized by liquefied gravity flows (Sassa and

Takagawa, 2019). Another study concludes that vertical ground displacement is the primary cause of the tsunami (Ulrich et al., 2019).

Our research will not focus on the factors that caused the tsunami. Instead, we will construct a mathematical model to calculate the maximum run-up height in Palu Bay. Since some previous research used calculations based on the assumption that the bay in question was a plane beach (Carrier et al., 2003; Chan and Liu, 2012; Didenkulova and Pelinovsky, 2019;

Drähne et al., 2016; Kanoğlu, 2004; Madsen and Fuhrman, 2007; Sammarco and Renzi, 2008; Synolakis, 1991), we will try to find a more precise approach using three different linearly-inclined channel types for estimating maximum run-up in Palu bay or other narrow bays: one with a parabolic cross-section, one with a triangular cross-section, and a plane beach. Henceforth, each of these will be referred to as the parabolic, triangular, and plane approximations. This idea has been used in previous papers (Choi et al., 2008; Didenkulova, 2013; Didenkulova et al., 2007; Didenkulova and Pelinovsky, 2011b).

We will start this research from formulating a mathematical model to describe the wave propagation in a bay using the generalized nonlinear shallow water equations. These equations depend on the cross-section of the bay that we choose. Since we use three different scenarios: the parabolic approximation, the triangular approximation, and the plane beach, we will also have three shallow water equations to be solved. Next, we will solve these equations analytically. The nonlinear shallow equations will be solved with the aid of the Riemann invariant and a hodograph transformation. It will allow us to have a

solvable linear partial differential equation. We can also find an expression for the physical variables in terms of the wave potential function. By assuming the incident wave is a sinusoidal wave, we will obtain a formulation for the maximum run-up height. After that, we have to determine two fixed points on the map: to observe the 2018 Palu tsunami wave run-up height on the shoreline, and to observe the tsunami wave properties in the body of water. We will use previous research to obtain the wave properties and the bathymetry map to obtain the water depth in the observed location. Substituting these numbers to the

maximum run-up height formulation, we will obtain a rough estimation of maximum run-up for each scenario of bay geometry. At last, we can compare the estimated run-up height with the observed run-up height. By calculating and comparing the error of each bay shape, we can find the most precise bay geometry to approximate a narrow and long bay–in this case, Palu Bay.


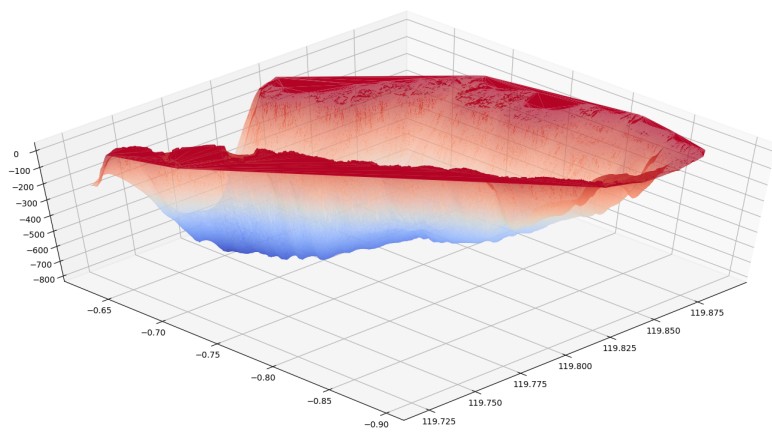

**Figure 1.** Bathymetry of Palu Bay

## 2   Mathematical Model

The tsunami run-up in the 2018 Palu tsunami can be analyzed in the framework of the nonlinear shallow water equations.
Since a nonlinear wave propagates with incompressible and inviscid water flow in a long and narrow bay, we will use the more
generalized nonlinear shallow water equations. The equations are,

$$\frac{\partial S}{\partial t} + \frac{\partial (Su)}{dx} = 0, \tag{1}$$

$$\frac{\partial u}{\partial t} + u\,\frac{\partial u}{\partial x} + g\,\frac{\partial H}{\partial x} = g\,\frac{dh}{dx}, \tag{2}$$

where $S(H)$ is the cross-section area of the bay, $H(x,t) = h(x) + \eta(x,t)$, $h(x)$, and $\eta(x,t)$ are respectively the perturbed water
depth, the unperturbed water depth, and the vertical displacement of the water surface along the main axis (the $x$-axis), $u(x,t)$
is the depth-averaged velocity over the cross-section, $g$ is gravitational acceleration, $x$ is coordinate directed onshore, and $t$ is
time. The longitudinal projection of the main axis and cross section of bay are shown in **Figure 2**

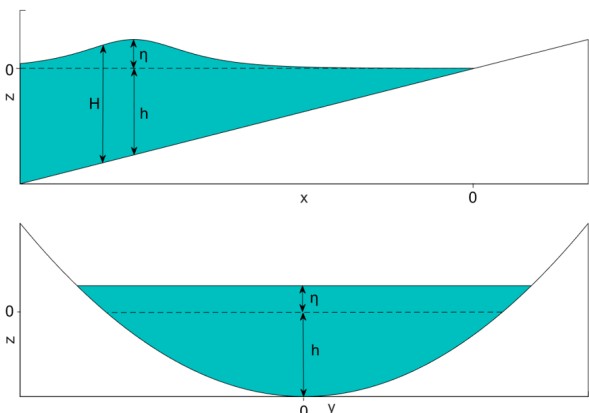

**Figure 2.** (a) Longitudinal projection of the main axis; (b) Cross-section of the bay

In order to find a more precise maximum run-up height estimation in the long and narrow Palu Bay analytically, we will assume the bay geometry satisfies three different scenarios of bay geometry: a linearly-inclined bay with a plane beach, a linearly-inclined bay with a parabolic bay shape, and a linearly-inclined bay with a triangular bay shape. Geometric model approximation of Palu Bay is given in **Figure 3**. The parabolic and triangular bay shape are respectively satisfying these proportional relationships,

$$z \propto |y|^2, \; z \propto |y|. \tag{3}$$

Accordingly, the cross-section area of these bay shapes, $S(H)$, can be described respectively as

$$S \propto H^{\frac{3}{2}}, \; S \propto H^2. \tag{4}$$

While the linearly inclined bay can be defined as

$$h(x) = -\alpha x, \tag{5}$$

where $\alpha$ is the constant angle of the bay's incline. As a result, we can acquire the governing equations for both cases of bay geometry. In the case of the parabolic approximation, the governing equations are

$$\frac{\partial H}{\partial t} + u\frac{\partial H}{\partial x} + \frac{2H}{3}\frac{\partial u}{\partial x} = 0, \tag{6}$$

$$\frac{\partial u}{\partial t} + u\frac{\partial u}{\partial x} + g\frac{\partial H}{\partial x} = -g\alpha, \tag{7}$$

while in the case of triangular bay, the governing equations are

$$\frac{\partial H}{\partial t} + u\frac{\partial H}{\partial x} + \frac{H}{2}\frac{\partial u}{\partial x} = 0, \tag{8}$$

$$\frac{\partial u}{\partial t} + u\frac{\partial u}{\partial x} + g\frac{\partial H}{\partial x} = -g\alpha. \tag{9}$$




These governing equations have similar forms with the classical shallow water equations. The distinction between these equations is the coefficient in front of the $H\partial u/\partial x$ term. The coefficient on the parabolic and triangular bay cases are respectively 2/3 and 1/2, whereas on the classical shallow water equations, for the plane beach, the coefficient is 1 (Carrier and Greenspan, 1958).

$$\frac{\partial H}{\partial t} + u\frac{\partial H}{\partial x} + H\frac{\partial u}{\partial x} = 0, \tag{10}$$

$$\frac{\partial u}{\partial t} + u\frac{\partial u}{\partial x} + g\frac{\partial H}{\partial x} = -g\alpha. \tag{11}$$

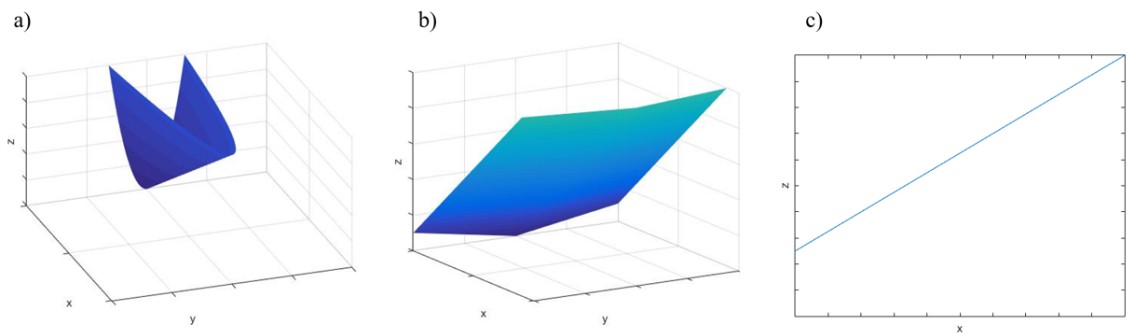

**Figure 3.** Geometric model approximation of Palu Bay; (a) Linearly-inclined channel with parabolic cross-section; (b) Linearly-inclined channel with triangular bay; (c) Plane beach

## 3   Analytical Solution

In order to solve the governing equations analytically, we will transform the system of nonlinear partial differential equations to a linear partial differential equation by using the Riemann invariant and a hodograph transformation. The corresponding Riemann invariant for the parabolic bay governing equations is described as

$$J_\pm = u \pm \sqrt{6gH} - gt\alpha. \tag{12}$$

Using this transformation, we can simplify the governing equations to a linear partial differential equation,

$$\frac{\partial^2 \Phi}{\partial \lambda^2} - \frac{\partial^2 \Phi}{\partial \sigma^2} - \frac{2}{\sigma}\frac{\partial \Phi}{\partial \sigma} = 0. \tag{13}$$

and the physical variables can be expressed in terms of wave potential function $\Phi(\sigma, \lambda)$,

$$x = \frac{1}{2g\alpha}\left(-\frac{\sigma^2}{3} + \frac{\partial \Phi}{\partial \lambda} - u^2\right), \tag{14}$$

$$u = \frac{3}{2\sigma}\frac{\partial \Phi}{\partial \sigma}, \tag{15}$$

$$\eta = \frac{1}{2g}\frac{\partial \Phi}{\partial \lambda} - \frac{u^2}{2g}. \tag{16}$$


The variables $\lambda$ and $\sigma$ represent generalized coordinates. The variable $\sigma$ is related to the perturbed water depth along the main axis, and $\sigma = 0$ corresponds to the moving shoreline.

In the case of the triangular approximation, we will use the same method as above. The corresponding Riemann invariant is
described as

$$J_{\pm} = u \pm \sqrt{8gH} - gt\alpha. \tag{17}$$

Furthermore, the governing equation can be transformed to a linear partial differential equation,

$$\frac{\partial^2 \Phi}{\partial \lambda^2} - \frac{\partial^2 \Phi}{\partial \sigma^2} - \frac{3}{\sigma}\frac{\partial \Phi}{\partial \sigma} = 0, \tag{18}$$

and the physical variables can be expressed as,

$$x = \frac{1}{2g\alpha}\left( -\frac{\sigma^2}{4} + \frac{\partial \Phi}{\partial \lambda} - u^2 \right), \tag{19}$$

$$u = \frac{2}{\sigma}\frac{\partial \Phi}{\partial \sigma}, \tag{20}$$

$$\eta = \frac{1}{2g}\frac{\partial \Phi}{\partial \lambda} - \frac{u^2}{2g}. \tag{21}$$

While for the plane approximation, the corresponding Riemann invariant is defined as

$$J_{\pm} = u \pm \sqrt{4gH} - gt\alpha. \tag{22}$$

The wave equation for the plane approximation is (Didenkulova et al., 2008; Pelinovsky and Mazova, 1992)

$$\frac{\partial^2 \Phi}{\partial \lambda^2} - \frac{\partial^2 \Phi}{\partial \sigma^2} - \frac{1}{\sigma}\frac{\partial \Phi}{\partial \sigma} = 0. \tag{23}$$

and the physical variables are

$$x = \frac{1}{2g\alpha}\left( -\frac{\sigma^2}{2} + \frac{\partial \Phi}{\partial \lambda} - u^2 \right), \tag{24}$$

$$u = \frac{1}{\sigma}\frac{\partial \Phi}{\partial \sigma}, \tag{25}$$

$$\eta = \frac{1}{2g}\frac{\partial \Phi}{\partial \lambda} - \frac{u^2}{2g}. \tag{26}$$

Assume the incident wave is a sinusoidal wave. Note that the incident wave potential function has to satisfy the equations (13), (18), and (23). The particular bounded solution for the equation (18) has the form

$$\Phi(\sigma, \lambda) = \frac{Q_0}{\sigma^{1/2}} J_{1/2}\left( \frac{\omega \sigma}{g\alpha} \right) \sin\left( \frac{\omega \lambda}{g\alpha} + \phi_0 \right), \tag{27}$$

whilst the bounded solution for the triangular approximation is

$$\Phi(\sigma, \lambda) = \frac{Q_1}{\sigma} J_1\left( \frac{\omega \sigma}{g\alpha} \right) \sin\left( \frac{\omega \lambda}{g\alpha} + \phi_1 \right), \tag{28}$$





and the bounded solution for the plane approximation is

$$\Phi(\sigma, \lambda) = Q_2 \, J_0\left(\frac{\omega\sigma}{g\alpha}\right) \sin\left(\frac{\omega\lambda}{g\alpha} + \phi_2\right), \tag{29}$$

where $J_n$ is the Bessel function of the first kind with order of $n$ and $Q_0, Q_2, Q_2, \phi_0, \phi_1, \phi_2$ are arbitrary constants which should

be determined. Using these potential wave functions and the corresponding physical variables, we can find an expression for

the run-up in the shore. The maximum run-up height on both bay shapes can be expressed as a function of the amplitude of the

wave at a fixed point $|x| = L$, $A$, and the multiplication of wave angular velocity, $\omega$ and wave travel time $\tau$. In the case of the

parabolic approximation, we can formulate the maximum run-up height as

$$R = 2A\omega\tau, \tag{30}$$

where the travel time, $\tau$, is defined as

$$\tau = \sqrt{\frac{6L}{g\alpha}}, \tag{31}$$

and in the case of triangular approximation, the maximum run-up height can be expressed as

$$R = \sqrt{\frac{\pi}{2}} A(\omega\tau)^{3/2}, \tag{32}$$

where the travel time, $\tau$, is defined as

$$\tau = \sqrt{\frac{8L}{g\alpha}}. \tag{33}$$

For the plane approximation, the maximum run-up height can be expressed as (Golinko et al., 2006; Rybkin et al., 2014),

$$R = A\sqrt{2\pi\omega\tau}, \tag{34}$$

and the travel time, $\tau$,

$$\tau = \sqrt{\frac{4L}{g\alpha}}. \tag{35}$$

The wave amplification, $R/A$, can be expressed as a function of $\omega\tau$. The wave amplification on the parabolic approximation is

directly proportional to $\omega\tau$, whereas in the case of the triangular approximation the wave amplification is directly proportional

to $\omega\tau$ to the power of three-halves, and on the plane approximation the wave amplification is directly proportional to $\omega\tau$ to

the power of one-half. For a small value of $\omega\tau$, the wave on the plane approximation tends to be amplified to a greater extent

than the wave on the parabolic triangular approximation. For a larger value of $\omega\tau$, the wave on the triangular approximation is

amplified to a greater extent than on the parabolic approximation, followed by the plane approximation. Therefore, it is proven

that the shape of the bay plays a large role in amplifying the incident wave. Nevertheless, we cannot guarantee that long and

narrow bays will amplify waves to greater heights than plane beaches, since the wave amplification factor still depends on the

angular frequency and travel time of the wave. These relationships are described in the figure below.

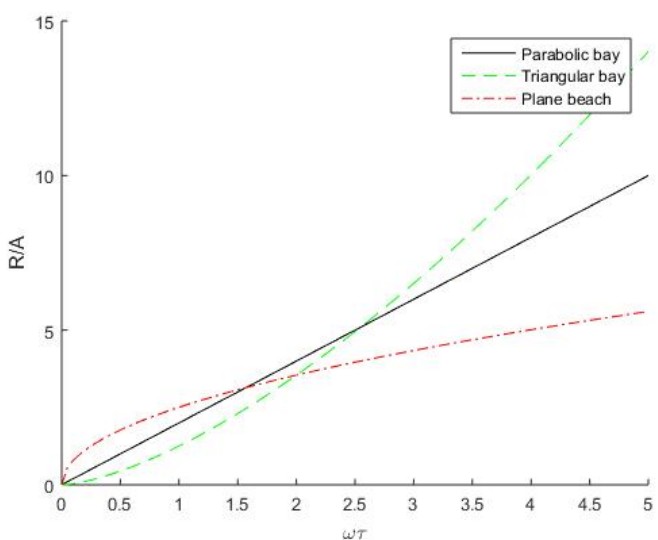

**Figure 4.** Wave amplification as a function of $\omega\tau$ for parabolic bay, triangular bay, and plane beach profile

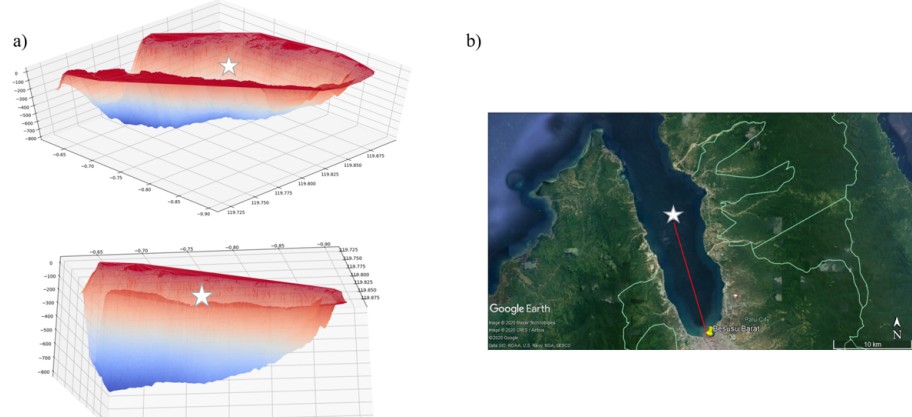

**Figure 5.** Fixed point location $(0°45'4.08"\,S, 119°49'19.4"\,E)$ on Palu Bay © Google Earth. The fixed point is shown as white star; (a) The location of the fixed point in the bathymethry of Palu Bay. It is located at the furthest part of linearly inclined bay; (b) The image showing the distance between the shore line (Besusu Barat) and the fixed point

In order to calculate the estimated maximum run-up height, we will determine a fixed point to represent the shoreline on the dry land. We choose Besusu Barat $(0°53'8.67"\,S, 119°51'44.42"\,E)$ as our point of observation since it is located at the bayhead and has the highest run-up height, 4.89 meters, compared to other observation points (Widiyanto et al., 2019). Next, we will choose the fixed point $|x| = L$ of the furthest part of the linearly inclined bay $(0°45'4.08"\,S, 119°49'19.4"\,E)$. The fixed point is shown on **Figure 5**. Therefore, the distance between the fixed point from the shoreline is $L = 15800$ meters.

According to the bay's bathymetry, the sea depth on the fixed point, $h_0$, is $\sim$700 meters. Using this information, we can calculate the bottom slope of the linearly inclined Palu bay,

$$\alpha = \frac{h_0}{L},$$ (36)

where $h_0$ is the depth of the fixed point and $L$ is the distance of the fixed point to the shoreline. Hence, the bottom slope of the bay is 0.0443. Since the angular frequency can be described as $\omega = \frac{2\pi}{T}$, therefore the maximum run up is inversely proportional to the period of the tsunami wave to a particular positive exponent. Observe that the maximum run up is proportional to the amplitude on the fixed point. Hence, the maximum amplitude and the minimum period of tsunami wave will be chosen to calculate the maximum estimated run-up height.

We will approximate the maximum amplitude of tsunami wave from the simulation released by the National Oceanic and Atmospheric Administration (NOAA). The simulation result is shown on **Figure 6**. We can estimate the amplitude on the fixed point as 0.15 meters. From previous research, we know that the period of the tsunami wave is 3 minutes (Heidarzadeh et al., 2019). The parameters of the tsunami wave recorded by local tide gauges are shown on Table 1.

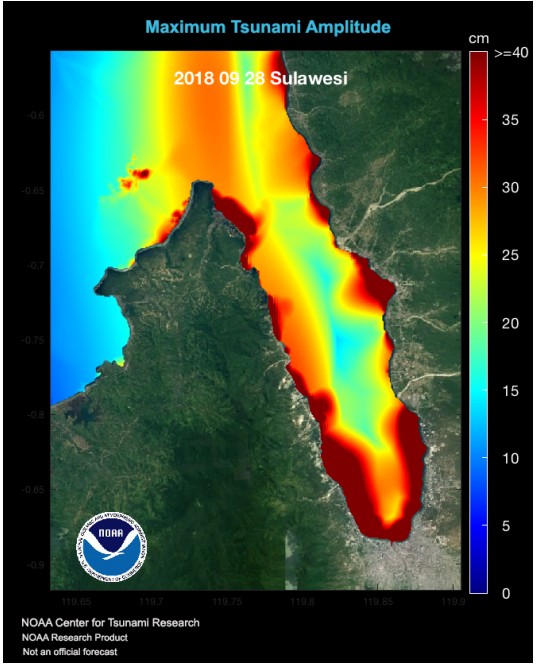

**Figure 6.** The simulation of tsunami wave maximum amplitude by NOAA Center for Tsunami Research, (NOAA, 2018)





**Table 1.** Parameters of the Sulawesi tsunami of 28 September 2018 recorded by local tide gauges (Main shock, $Mw$ = 7.5 at 10:02:43 UTC) (Heidarzadeh et al., 2019)

| Station | First Wave | | | Max Wave | | | | |
| | Arrival time (UTC) | Travel time | Amplitude (cm) Sign | Max amplitude (cm) | Time (UTC) of max amplitude | Max wave height | Duration high-energy waves (h) | Visible period (min) |
| --- | --- | --- | --- | --- | --- | --- | --- | --- |
| Pantoloan | 10:07 | 5 min | - 203.4 | 176.4 | 10:10 | 379.8 | 5.5 | 3-4 |
| Mamuju | 10:21 | 19 min | - 6.9 | 14.3 | 10:57 | 24.2 | > 14 | 10-12 |

Using these data, we can simply calculate the maximum run-up height on Besusu Barat, Palu with the plane, parabolic and triangular approximations we have described. The maximum run-up height in the case of parabolic bay is 4.89 meters. In the case of triangular bay, the prediction for maximum run-up height is 15.35 meters. While in the case of plane beach, the maximum run-up height reached at most 1.37 meters.

**Table 2.** Result

| Observed location | Latitude | Longitude | Recorded height (m) | Parabolic bay | | Triangular bay | | Plane beach | |
| | | | | Estimated height (m) | Error[a] (m) | Estimated height (m) | Error (m) | Estimated height (m) | Error (m) |
| --- | --- | --- | --- | --- | --- | --- | --- | --- | --- |
| Besusu Barat | $0°53'8.67"S$ | $119°51'44.42"E$ | 4.89 | 4.89 | 0 | 15.35 | +10.46[b] | 1.37 | -3.52 |

[a] Error is calculated by subtracting the recorded height from the estimated height.

[b] Positive sign and negative sign represent an overestimate and an underestimate, respectively, relative to the recorded height.

## 4   Conclusions

Solving the generalized shallow water equations analytically allows us to estimate the maximum run-up height on various bay types. The maximum run-up in Besusu Barat, Palu on the 2018 Palu tsunami is studied using this method by approximating the bay geometry with three different scenarios: the parabolic bay, the triangular bay, and the plane beach. The recorded run-up height is 4.89 meters. Meanwhile, the maximum run-up heights over the three approximations for Palu Bay's geometry are as follows: 1.37 meters for a plane beach, 4.89 meters for a linearly-inclined channel of parabolic cross-section, and 15.35 me-

ters for a linearly-inclined channel of triangle cross-section. The approximation with a parabolic cross-section yields a better estimate of run-up height than both the plane beach and linearly-inclined channel of triangle cross-section, since the run-up height calculated using the parabolic approximation is the same as the recorded height whereas the figure obtained using the plane approximation underestimates the run-up height by 3.52 meters and the figure obtained using the triangular approximation overestimate it by 10.46 meters. The percentage errors of the plane beach and triangular beach are 71.98% and 297.16%,

respectively.





It is suggested to use this direct formulation in order to obtain an instant forecast of the maximum tsunami wave run-up in a narrow bay, since the formulation only depends on four variables: the maximum amplitude and the minimum period of the tsunami wave, the sea depth at the fixed point, and the distance from the fixed point to the shoreline. This can help government

agencies save computation time. It is also suggested to use the parabolic bay approach rather than the plane beach approach to improve the tsunami early warning systems for predicting run-up on the narrow bays. Using the parabolic formulation, we can have a more precise estimation of the maximum run-up height in Palu Bay or other narrow bays. Hence, with a better understanding of this tragic event, we can reduce the number of casualties and damages in the upcoming disaster.

*Data availability.* The parameters of Sulawesi tsunami of 28 September 2018 were obtained from National Oceanic and Atmospheric Ad-

ministration (NOAA). The tide gauge records were obtained from Heidarzadeh et al. (2019) and provided by the Agency for Geospatial Information, Indonesia (BIG).

*Author contributions.* All authors worked on the conceptualization of this paper. Ikha Magdalena provided supervision and the data set. Antonio Hugo Respati Dewabrata prepared the manuscript and conducted the formal analysis. Alvedian Mauditra Aulia Matin reviewed and edited the manuscript. Adeline Clarissa contributed to visualization and designs. Muhammad Alif Aqsha supported the preparation, creation,

and presentation of the study.

*Competing interests.* The authors declare that they have no conflict of interest.





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
