# Peer review of "A mathematical formulation for estimating maximum run-up height of 2018 Palu tsunami"

_Natural Hazards and Earth System Sciences, 2020_

## Referee Comment (RC1) · Anonymous Referee #1 · 13 Aug 2020

<General comment>

The paper deals with tsunami amplification in a closed bay. The authors applied an analytical model to the case of the 2018 Palu Tsunami and claimed that the model described the tsunami run-up in a narrow bay well. The analytical model itself is an existing one and I do not see anything new in it. The way of applying the model to the case appears to be rough, and thus, the results are not convincing. I do not think the paper is in publishable quality. The authors need to carefully discuss the model applicability and validate the model from different perspectives.

<Specific comments>

(1) The bay topography is shown by a 3D plot in an ambiguous way (Figure 1). I suggest

the authors to provide the longitudinal cross-section and transversal cross-sections at some representative locations with a length scale (not in longitude/latitude). Additionally, the authors need to compare the longitudinal transition of the cross-sectional area in comparison to the idealized bays in the model, which is important in discussions of the wave funneling effect.

(2) Please describe the general characteristics of the tsunami in Palu bay and justify the use of the section-averaged shallow-water model prior to the model introduction. The model is based on some assumptions that hold for specific bay geometries relative to wavelength (e.g. shallow-water and narrow-bay assumptions).

(3) The analytical model is introduced separately for different bay types: rectangular (plane beach), parabolic, and triangular bays. But they can be derived in a unified manner using a single geometric parameter. This is recommended not only for the preciseness of the paper. It enables the authors to consider, for example, an idealized bay between the rectangular (plane beach) and parabolic bays for a better fit to the real bay bathymetry. See [1] for the general expression of the model.

(4) The run-up formulas presented in Page 7 are based on asymptotic approximations of the Bessel functions that appear in (27)-(29). As a result, the runup amplification factor (R/A) in Figure 4 goes to zero for a very long wave because the approximation does not work in the range of small omega*t. The authors need to clarify this and show that the Palu case is not in this range. The run-up formulas without the asymptotic approximation are given in [2].

(5) The model gives run-up height in an equilibrium state under a monochromatic wave. The actual tsunami is a transient one. The authors need to discuss the model applicability in this regard. If there are available tide records at different locations in the bay as shown in Table 1, the authors can validate the model from a different perspective other than the maximum run-up height at the bay head which might be affected by very short waves and local topography.

(6) The numerical result in Figure 6 exhibits strong transverse variations of the highest crest level in the bay indicating significant effects of transverse flows. This suggests that the tsunami propagated in the bay not as a plane wave as assumed in the model. The authors need to discuss the model applicability in this regard and associated limitations. See [2] for the effect of transverse flows in a closed bay.

(7) I do not deny the possibility that the model result agrees with the observed value at 10^-2 m accuracy as presented in the paper. It can happen by coincidence. But the authors need to carefully justify the model inputs since the agreement level is far beyond the model capability.

(8) In relation to the previous comment, the authors need to provide a basis for the choice of the input wave amplitude at the midpoint of the bay from the numerical result. There is a considerable degree of arbitrariness. The numerical result in Figure 6 gives a variation of the highest crest level within the bay, which was not necessarily produced by the same wave that caused the maximum run-up at the bayhead.

<References>

[1] Zahibo, N., Pelinovsky, E., Golinko, V. & Osipenko, N. 2006 Tsunami wave runup on coasts of narrow bays. Intl J. Fluid Mech. Res. 33, 106–118.

[2] Shimozono, T. (2016). Long wave propagation and run-up in converging bays. Journal of Fluid Mechanics, 798, 457-484.

---

## Author Comment (AC1) · 25 Aug 2020

Dear Referee #1,

First of all, thank you very much for all the suggestions! We would like to discuss some of the suggestions and comments. The responses of the suggestion and comment are as follows:

**General comments**: The paper deals with tsunami amplification in a closed bay. The authors applied an analytical model to the case of the 2018 Palu Tsunami and claimed that the model described the tsunami run-up in a narrow bay well. The analytical model itself is an existing one and I do not see anything new in it. The way of applying the model to the case appears to be rough, and thus, the results are not convincing. I do not think the paper is in publishable quality. The authors need to carefully discuss the model applicability and validate the model from different perspectives.

**Response:** We indeed used the existing model to estimate the maximum run-up height. This is why we formulate this paper as a brief communication, not a research article. We want to emphasize that we were looking for a simple formulation to find the maximum run-up, which the observed run-up height will not exceed. It is only used to calculate the run-up of the tsunami in the worst-case scenario through this simple calculation. We hope that the field observer can easily and quickly estimate the maximum run-up of a tsunami in a certain location without using a time-consuming numerical simulation and a large dataset.

**Specific comment 1:** The bay topography is shown by a 3D plot in an ambiguous way (Figure 1). I suggest the authors to provide the longitudinal cross-section and transversal cross-sections at some representative locations with a length scale (not in longitude/latitude). Additionally, the authors need to compare the longitudinal transition of the cross-sectional area in comparison to the idealized bays in the model, which is important in discussions of the wave funneling effect.

**Response:** We have already prepared both longitudinal and transversal cross-sections for the next revision. We neglect the wave funneling effect since we are focusing on the worst scenario along the main axis toward the bayhead (shoreline). We will explain further in the next response.

[Figure]

**Figure 1. Bathymetry of the Palu Bay**

**Specific comment 2:** Please describe the general characteristics of the tsunami in Palu bay and justify the use of the section-averaged shallow-water model prior to the model introduction. The model is based on some assumptions that hold for specific bay geometries relative to wavelength (e.g. shallow-water and narrow-bay assumptions).

**Response:** We have checked and proved that the general characteristics in Palu bay satisfy the assumption of shallow-water model. We intend to add these in our next revised introduction.

"*The period of the tsunami in Palu is approximately 3 minutes [2]. According to The National Agency for Disaster Countermeasure (BNPB) [1,3,4], the velocity of the tsunami is at least 800 kilometers per hour. We can find the wavelength of the tsunami from these variables, lambda = 40 km. The ratio between the depth of the Palu bay and the wavelength is less than 0.05. Therefore, the tsunami in Palu bay can be treated as a shallow-water wave.*"

**Specific comment 3 :** The analytical model is introduced separately for different bay types: rectangular (plane beach), parabolic, and triangular bays. But they can be derived in a unified manner using a single geometric parameter. This is recommended not only for the preciseness of the paper. It enables the authors to consider, for example, an idealized bay between the rectangular (plane beach) and parabolic bays for a better fit to the real bay bathymetry. See [1] for the general expression of the model.

**Response:** It is a really nice suggestion. However, we wanted to emphasize each geometry using the step-by-step elaboration.

**Specific comment 4:** The run-up formulas presented in Page 7 are based on asymptotic approximations of the Bessel functions that appear in (27)-(29). As a result, the runup amplification factor (R/A) in Figure 4 goes to zero for a very long wave because the approximation does not work in the range of small omega*t. The authors need to clarify this

and show that the Palu case is not in this range. The run-up formulas without the asymptotic approximation are given in [2].

**Response:** We have checked the angular frequency and travel time of the tsunami in Palu bay. The multiplications of these variables for each bay geometry are large enough to be approximated by the Bessel function asymptotic approximation. We intend to add this to our revised paper.

"*The travel time of tsunami waves in the plane beach, parabolic, and triangular bays, respectively, are 6.3555, 7.7839, and 8.9881 minutes. The angular frequency of the wave is 0.0349 s^-1. Hence, the multiplications of the travel time and angular frequency for each bay geometries, respectively, are 13.3110, 16.3026, and 18.8246. The differences between the Bessel functions and asymptotic Bessel functions approximation for these values in each bay geometric case are small enough (less than 10^-2).*"

**Specific comment 5:** The model gives run-up height in an equilibrium state under a monochromatic wave. The actual tsunami is a transient one. The authors need to discuss the model applicability in this regard. If there are available tide records at different locations in the bay as shown in Table 1, the authors can validate the model from a different perspective other than the maximum run-up height at the bay head which might be affected by very short waves and local topography.

**Response:** In this project, our intention is to calculate the worst case of the tsunami maximum run-up height along the main axis. The monochromatic wave travels at a constant energy compared to the transient wave. The wave energy is proportional to the squared of the amplitude. Hence, assigning the maximum amplitude of the transient wave as the constant amplitude of the sinusoidal wave will capture the worst case scenario that we wish to anticipate.

**Specific comment 6:** The numerical result in Figure 6 exhibits strong transverse variations of the highest crest level in the bay indicating significant effects of transverse flows. This suggests that the tsunami propagated in the bay not as a plane wave as assumed in the model. The authors need to discuss the model applicability in this regard and associated limitations. See [2] for the effect of transverse flows in a closed bay.

**Specific comment 7:** I do not deny the possibility that the model result agrees with the observed value at $10^-2$ m accuracy as presented in the paper. It can happen by coincidence. But the authors need to carefully justify the model inputs since the agreement level is far beyond the model capability.

**Response:** We chose the model inputs in accordance with our goal, which was to look for the maximum run-up height in the tsunami's worst-case scenario. We were looking for the worst case of the tsunami maximum run-up height. We took into account the fact that the monochromatic plane wave transfers a constant amount of energy. On the other hand, the transverse wave breaking dissipation limited the energy transfer to the bay head region in many bays, as said by Shimozono [2].) Since we only focused on looking at the tsunami's maximum run-up height, we assigned the extreme value of each parameter to the formulation. The models that give the same or overestimated result are better than the model that gives underestimated results. This is the reason why we do not state that the parabolic bay shape is better than the triangular bay shape. We tried to apply this formulation on other tsunami cases and it works well enough (the model provides an overestimation). In this research, we

only want to emphasize the Palu tsunami, which was why we didn't show the other cases results.

**Specific comment 8:** In relation to the previous comment, the authors need to provide a basis for the choice of the input wave amplitude at the midpoint of the bay from the numerical result. There is a considerable degree of arbitrariness. The numerical result in Figure 6 gives a variation of the highest crest level within the bay, which was not necessarily produced by the same wave that caused the maximum run-up at the bayhead.

**Response:** With the aim of obtaining the input wave amplitude, first of all we drew the main axis from the shoreline to the mouth of the bay. Afterwards, we compared the bathymetry of the Palu and picked the deepest point of the monotonically-decreasing part of the sea profile along the main axis from the shoreline. Finally, we approximated the value of the amplitude by plotting the coordinate of this numerical result and choosing the nearest tenth value of the amplitude from the numerical result.

According to the chosen model, the monochromatic wave run-up is proportional to the observed amplitude. As the previous response, the monochromatic wave travels at constant energy, this will lead to the worst-case scenario of the tsunami. If we are using the maximum amplitude on this model formulation, it will produce the maximum run-up on the shoreline.

**REFERENCES**

[1] Al Jazeera and News Agencies (2018), 'Indonesia earthquake and tsunami: All the latest updates', Al Jazeera, 8 October. Available at: https://www.aljazeera.com/news/2018/10/indonesia-earthquake-tsunami-latest-updates-181003060041729.html (Accessed: 16 August 2020)..

[2] Heidarzadeh, M., Muhari, A. and Wijanarto, A.B., 2019. Insights on the source of the 28 September 2018 Sulawesi tsunami, Indonesia based on spectral analyses and numerical simulations. Pure and Applied Geophysics, 176(1), pp.25-43.

[3] La Chaîne Info (2018), 'Tsunami en Indonésie : pourquoi une vague de 1,50 m a-t-elle été aussi dévastatrice ?', La Chaîne Info, 29 September. Available at: https://www.lci.fr/international/tsunami-en-indonesie-celebes-sulawesi-pourquoi-une-vague-de-1-50-m-a-t-elle-ete-aussi-devastatrice-2099917.html (Accessed: 16 August 2020)

[4] Savitri, Eva (2018), 'BNPB: Kecepatan Tsunami Palu 800 Km/Jam, Hancurkan Infrastruktur', DetikNews, 28 September. Available at: https://news.detik.com/berita/d-4234574/bnpb-kecepatan-tsunami-palu-800-kmjam-hancurkan-infrastruktur (Accessed: 16 August 2020)

---

## Referee Comment (RC2) · Anonymous Referee #2 · 10 Sep 2020

Dear authors,

It was interesting to read your manuscript and I am glad someone was trying to apply cross-sectionally averaged shallow water equations to model tsunamis the Palu bay. Especially, since the Palu bay has an almost ideal parabolic shape with a linear slope.

1) After reading the manuscript, I started to be significantly concerned regarding the applicability of cross-sectionally averages theory to the case of 2018 Sulawesi tsunami. The earthquake rupture happened not across the bay but rather diagonally and near its head. This could be easily seen e.g. at the NOAA tsunami modeling web-page (figure 6 is likely from there) https://nctr.pmel.noaa.gov/sulawesi20180928/ and https://www.youtube.com/watch?v=98scC02hNzo&feature=youtu.be See the attachement, figure 1, for the tsunami wave height right at its initiation. The cross-sectionally

averaged shallow water equations assume a uniform wave across the entire bay, which is not the case here.

2) In the manuscript it was mentioned that the period of the wave was 3 minutes. Besides the tectonic tsunami caused by the ocean bottom, there were several landslides along the lateral shores, and it is not clear whether the 3 minutes and due to the tectonic and/or landslide components. Overall, the paper would greatly benefit from a short discussion of the tsunami source, wave propagation, observations, etc. This can (or maybe cannot) put a solid footing for the cross-sectionally averaged theory chosen here to model runup.

3) Please explain applicability of the monochromatic wave assumption to compute the runup of Sulawesi tsunami in the Palu bay. This methodology is applicable in some theoretical computations to show importance of the bay geometry, wave period, etc. But we rarely have seen monochromatic tsunami waves. I would almost say 'never have' seen.

4) Derivations of the analytical solutions could be shortened since the bay profiles depend on the exponential. For example, a reader could be referred to Garayshin et al., (2016) and Anderson et al., (2017) who considered runup of long wave runup in the general case of U-shaped and V-shaped bays. These authors also showed that the triangular shaped bay can produce a larger runup, given all other parameters constant.

Garayshin, V., Harris, M., Nicolsky, D., Pelinovsky, E., & Rybkin, A. (2016). An analytical and numerical study of long wave runup in U-shaped and V-shaped bays. Applied Mathematics and Computation, 297, 187–197 Anderson, D., Harris, M., Hartle, H. et al. Run-Up of Long Waves in Piecewise Sloping U-Shaped Bays. Pure Appl. Geophys. 174, 3185–3207 (2017). https://doi.org/10.1007/s00024-017-1476-3

**Fig. 1.** first seconds of the modeled tsunami according to the PMEL NOAA simulations

---

## Author Comment (AC2) · 5 Oct 2020

Dear Referee #2,

Thank you for the time and effort that the reviewers have dedicated to provide your valuable feedback on our manuscript. The suggestions offered by the reviewers have been immensely helpful.

We have been able to incorporate changes to reflect most of the suggestions provided by the reviewers. We have responded to them individually, indicating exactly how we addressed each concern or problem and describing the changes we have made. The revisions have been approved by all authors.

Comment 1: After reading the manuscript, I started to be significantly concerned regarding the applicability of cross-sectionally averages theory to the case of 2018 Su-lawesi tsunami. The earthquake rupture happened not across the bay but rather diago-nally and near its head. This could be easily seen e.g. at the NOAA tsunami modeling webpage (figure 6 is likely from there) https://nctr.pmel.noaa.gov/sulawesi20180928/ and https://www.youtube.com/watch?v=98scC02hNzo&feature=youtu.be See the at-tachment, figure 1, for the tsunami wave height right at its initiation. The cross-sectionally averaged shallow water equations assume a uniform wave across the entire bay, which is not the case here.

Response: We agree that the earthquake rupture happened not across the bay but rather diagonally and near its head. According to Widiyanto, 2019 [4], there were three main tsunami waves that reached the beach. The first wave is less than 1 meter tall and the second and third waves are the more devastating tsunami waves. From the simulation, we can see that the diagonal Palu-Koro fault earthquake rupture initiated the transverse wave along the bay. This transverse wave struck our point of observation, Besusu Barat, for the first time at the eighth until the twelfth minute of the simulation. This first strike can be treated as the first wave. We can also see from the simulation that there was a wave created behind our fixed point, far from the shoreline near the Pantoloan city, with a high amplitude. Afterward, this wave propagated near-uniformly across the entire bay towards the bay mouth and struck the shore with a high amplitude at the seventeenth until the twenty-fourth minute of the simulation. Please kindly see the Figure 1 below. Because of this reason, we can assume that this wave was the second or third tsunami. Since we are calculating the upper boundary of the run-up, we did not calculate the propagation from the initial earthquake rupture moment, when the first wave is more likely to happen. However, we were calculating the propagation when the near-uniformly wave propagated across the bay towards the onshore and causing a devastating event. This is why we are able to use the cross-sectionally averaged shallow water equations.

Comment 2: In the manuscript it was mentioned that the period of the wave was 3

minutes. Besides the tectonic tsunami caused by the ocean bottom, there were several landslides along the lateral shores, and it is not clear whether the 3 minutes and due to the tectonic and/or landslide components. Overall, the paper would greatly benefit from a short discussion of the tsunami source, wave propagation, observations, etc. This can (or maybe cannot) put a solid footing for the cross-sectionally averaged theory chosen here to model runup. Response: The 3-4 minutes period is the dominant wave period as recorded in de-tided sea level records as stated by Heidarzadeh, 2019 [2]. The dominant wave period is the wave period associated with the highest energetic waves in the area. Using the spectral analysis, it is suggested that the dominant period is the local tectonic source. It is also suggested that a small submarine landslide is occurred using the combination of spectral analysis, numerical analysis and field data [2].

It is a really nice suggestion. We are planning to add a short discussion of wave propagation and observations in Palu bay as in the previous reply.

"Three tsunami waves were recorded across the shore. The first wave is less than 1 meter and the second and third waves are the more devastating tsunami waves (Wahyu Widiyanto et al. (2019)). The dominant period of the wave is 3-4 minutes and it is suggested that this number is the result of a submarine landslide (Heidarzadeh et al. (2019)). From the NOAA simulation, the diagonal Palu-Koro fault earthquake rupture generated a transverse wave along the bay. This transverse wave struck our point of observation, Besusu Barat, for the first time at the eighth until the twelfth minute of the simulation. This first strike can be treated as the first wave. Also from the simulation, there was a sudden wave created behind our fixed point, far from the shoreline near the Pantoloan city, with a high amplitude. This nearly uniform wave propagated across the entire bay towards the bay mouth and struck the shore with a high amplitude at the seventeenth until the twenty-fourth minute of the simulation. It can be assumed that this wave was the second or third tsunami. Since we intended to calculate the upper boundary of the run-up, the initial earthquake rupture moment will not be the initial

condition of our calculation. However, we were only calculating the propagation when the wave propagated near-uniformly across the bay towards the onshore and causing a devastating event. Therefore, the cross-sectionally averaged shallow water equations will be used in our analytical analysis derivation."

Comment 3: Please explain applicability of the monochromatic wave assumption to compute the runup of Sulawesi tsunami in the Palu bay. This methodology is applicable in some theoretical computations to show importance of the bay geometry, wave period, etc. But we rarely have seen monochromatic tsunami waves. I would almost say 'never have' seen. Response: The aim of this paper is to find the upper boundary of the tsunami run-up height in the observed location. The monochromatic wave leads to the worst scenario of the tsunami run-up. It is easy to see that the monochromatic wave can lead to a more devastating run-up compared to the solitary wave, the usual approximation of a tsunami wave. The greater result of monochromatic wave run-up compared to solitary wave run-up in the parabolic bay can be also seen in [1].

Comment 4: Derivations of the analytical solutions could be shortened since the bay profiles depend on the exponential. For example, a reader could be referred to Garayshin et al., (2016) and Anderson et al., (2017) who considered run-up of long wave run-up in the general case of U-shaped and V-shaped bays. These authors also showed that the triangular shaped bay can produce a larger run-up, given all other parameters constant. Response: It is a really nice suggestion. However, the reason we elaborate on the formulae derivation is we want to emphasize each characteristic of the bay profile.

References [1] Golinko, V., Osipenko, N., Pelinovsky, E.N. and Zahibo, N., 2006. Tsunami wave runup on coasts of narrow bays. International Journal of Fluid Mechanics Research, 33(1). [2] Heidarzadeh, M., Muhari, A. and Wijanarto, A.B., 2019. Insights on the source of the 28 September 2018 Sulawesi tsunami, Indonesia based on spectral analyses and numerical simulations. Pure and Applied Geophysics, 176(1), pp.25-43. [3] NOAA: Modeled Amplitudes for Sulawesi Tsunami 2018 September

28, MOST Forecast Model, 2018 [4] Widiyanto, W., Santoso, P.B., Hsiao, S.C. and Imananta, R.T., 2019. Post-event field survey of 28 September 2018 Sulawesi earthquake and tsunami. Nat. Hazards Earth Syst. Sci, 1, pp.1-23.

[Figure]

[Figure]

Simulation time: 17 m 01 s    Simulation time: 19 m 19 s    Simulation time: 24 m 22 s

**Figure 1. Nearly uniform tsunami wave propagation towards the bay mouth
(NOAA, 2019) [3]**

**Fig. 1.**